# SARS-CoV-2 Omicron variant is attenuated for replication in a polarized human lung epithelial cell model

Christin Mache[1], Jessica Schulze[1], Gudrun Holland[2], Daniel Bourquain [3], Jean-Marc Gensch[1], Djin-Ye Oh[1], Andreas Nitsche[3], Ralf Dürrwald [1], Michael Laue [2] & Thorsten Wolff [1✉]

SARS-CoV-2 and its emerging variants of concern remain a major threat for global health. Here we introduce an infection model based upon polarized human Alveolar Epithelial Lentivirus immortalized (hAELVi) cells grown at the air–liquid interface to estimate replication and epidemic potential of respiratory viruses in the human lower respiratory tract. hAELVI cultures are highly permissive for different human coronaviruses and seasonal influenza A virus and upregulate various mediators following virus infection. Our analysis revealed a significantly reduced capacity of SARS-CoV-2 Omicron BA.1 and BA.2 variants to propagate in this human model compared to earlier D614G and Delta variants, which extends early risk assessments from epidemiological and animal studies suggesting a reduced pathogenicity of Omicron.

[1] Influenza and other Respiratory Viruses (Unit 17), Department of Infectious Diseases, Robert Koch Institute, Seestraße 10, 13353 Berlin, Germany. [2] Advanced Light and Electron Microscopy (ZBS 4), Centre for Biological Threats and Special Pathogens, Robert Koch Institute, Seestraße 10, 13353 Berlin, Germany. [3] Highly Pathogenic Viruses (ZBS 1), Centre for Biological Threats and Special Pathogens, Robert Koch Institute, Seestraße 10, 13353 Berlin, Germany. ✉email: wolfft@rki.de

Emergence of the human Severe Acute Respiratory Syndrome Coronavirus 2 (SARS-CoV-2) in 2020 lead to the global Coronavirus Disease 2019 (COVID-19) pandemic, which caused so far more than 6.4 million deaths worldwide. SARS-CoV-2 is mainly transmitted via virus-containing droplets and aerosols[1]. Acute COVID-19 can be divided into up to three stages of disease with increasing severity including the early phase, in which the virus establishes infection in tissues of the upper respiratory tract, which is often associated with no or mild symptoms (stage I). During stage II (pulmonary phase) active viral replication in cells of distal airways can trigger pneumonia including localized inflammation in the lung and hypoxia. In some cases there is progression to even more severe disease such as Acute Respiratory Distress Syndrome (ARDS), shock and cardiac failure as a consequence of systemic hyperinflammation (stage III)[2]. However, the virus may infect the lung also directly after inhalation of virus-containing aerosols[3].

Since late 2020, several viral variants of concern (VOC) including Alpha, Beta, Gamma, and Delta started to emerge with some of them becoming predominant within a short period of time in distinct geographical regions leading to strong waves of disease[4]. VOCs pose a serious risk to global public health due to an increased transmissibility and escape from adaptive immune responses, which is caused by distinct genotypical changes leading to variations in the viral Spike and other proteins. The most recent VOC termed Omicron, which emerged in countries of Southern Africa in November 2021 and is further divided into sublineages BA.1 to BA.5, encodes more than 25 amino acid changes, several small deletions (BA.1: amino acids 69–70, 143–145, 211; BA.2: amino acids 24–26) and 1 insertion in the spike of BA.1 at position 214[5,6]. While the transmission in unvaccinated individuals appears to be comparable to the previously dominating Delta, the Omicron VOC shows immune evasion in fully vaccinated and even booster-vaccinated people, which possibly explains the rapid spread to all continents[7]. From South Africa, it was reported that the Omicron VOC is more likely to infect younger people and cause milder disease[8]. However, it is less well understood what courses of the disease occur in the elderly and other vulnerable groups with increased risk for severe COVID-19.

Severe COVID-19 is characterized by persisting high-level viral shedding in the lower respiratory tract[9]. To assess how emerging respiratory viruses including SARS-CoV-2 variants differ regarding their potential to propagate in the human lower respiratory tract and thereby induce severe disease, reliable infection models mimicking viral replication and host cell activation in patients as closely as possible are highly desirable. Autopsy studies showed that SARS-CoV-2 infects type I and type II pneumocytes in the lung[10], which are the main cell types building the alveolar epithelium that forms a polarized and tight barrier facilitating fluid homeostasis and oxygen uptake through underlying blood capillaries in the alveolar space. Consistently, lung cells were shown to express SARS-CoV-2 host factors including the receptor angiotensin-converting enzyme 2 (ACE2) and transmembrane protease serine subtype 2 (TMPRSS2). Importantly, primary alveolar cells cultured at the air–liquid-interface (ALI) can recapitulate many facets of the alveolar epithelium partially, but quickly lose their in vivo phenotype and are poorly permissive for SARS-CoV-2 infection[10,11]. Several lung derived cell lines are currently in use to investigate SARS-CoV-2 infections, but many of them model the alveolar epithelium imperfectly, either because they lack the ability to form tight junctions (e.g., A549 or NCI-H1299 lines), or are of bronchial origin (e.g., Calu-3, NCI-H441, 16HBE14). Most of them are derived from tumors and may have deregulated homeostasis, in addition. Furthermore, lung organoids derived from primary human lung-specific progenitor or stem cells[12], or explanted human lung tissue have been used to study SARS-CoV-2 infections. The latter model replicates wild-type SARS-CoV-2 only to low levels of around $1 \times 10^4$ TCID$_{50}$[11] and both systems require prior ethical approval[12].

Here, we report on a model in polarized human Alveolar Epithelial Lentivirus immortalized (hAELVi) cell cultures[13] to study severe infections of SARS-CoV-2 and other respiratory viruses in the lower human respiratory tract. The immortalized hAELVi cells were initially derived from type I pneumocytes that cover the vast majority of the lung surface and which were immortalized using a lentiviral gene library[13]. Using filter supports, hAELVi cells can be differentiated under ALI conditions to a stratified, polarized epithelial cell layer with high barrier functions. This appears to be advantageous as respiratory cells polarized under ALI conditions resemble authentic body cells in their gene expression patterns more closely compared to cells grown in submerged cultures[14]. The hAELVi cells are commercially available and many aspects of their structure, physiology and responses to exterior stimuli have been characterized, but to the best of our knowledge it is not known whether they are suitable for the study of viruses.

## Results

**Polarized hAELVi ALI cultures express SARS-CoV-2 host factors.** hAELVi cells were seeded and grown on permeable filter inserts under liquid–liquid conditions for 72 h, after which cells were further incubated for up to 28 days under ALI (Fig. 1a). Cell cultures started to develop transepithelial electrical resistance (TEER) after 10 days under ALI with a high maximum value of 6000 $\Omega * cm^2$ approximately 21 days after seeding. Similar dynamics were observed for the permeability of the paracellular transport marker sodium fluorescein, which progressively decreased through day 14 indicating the presence of a tight epithelium (Fig. 1b).

Cells grown under ALI conditions adopt a rather cubic shape with an in many regions multilayered organization, and develop into a pseudostratified or stratified columnar epithelium at day 21 (Fig. 1c and Supplemental Fig. 1). The epithelial cells revealed many short apical microvilli (Supplemental Fig. 2a), cilia or their basal bodies could be detected exceptionally in single cells (Supplemental Fig. 2b). Multilamellar bodies, which are typical for alveolar type II cells, were not detected. In summary, the cell layer at later stages morphologically resembled epithelium of the terminal bronchiolar segments of the lower respiratory tract, including also non-ciliated Clara cells.

Immunoblot and ELISA analyses showed that hAELVi cells upregulated expression of the ACE2 receptor and the host protease TMPRSS2 through the 28 days period grown under ALI conditions (Fig. 1d, e). In contrast, expression of the MERS-CoV receptor protein DPP4 decreased during the polarization process (Fig. 1e).

**hAELVi ALI cultures are a highly productive infection model for various respiratory viruses.** We proceeded to investigate the hAELVi cultures as hosts for the three human coronaviruses (CoV) SARS-CoV, MERS-CoV and an early SARS-CoV-2 from 2020 (D614G), as well as seasonal influenza A virus (IAV). Initially, virus propagation was analyzed on submerged hAELVi cell cultures, which revealed moderate replication capacity for all three coronaviruses growing to titers of between $2 \times 10^4$ pfu/ml and $1 \times 10^5$ pfu/ml, whereas IAV replicated to higher titers of around $1 \times 10^7$ pfu/ml (Fig. 2a). Interestingly, growth curve analyses of progeny viruses harvested in apical washes of the ALI cultures revealed an increased productivity of the polarized

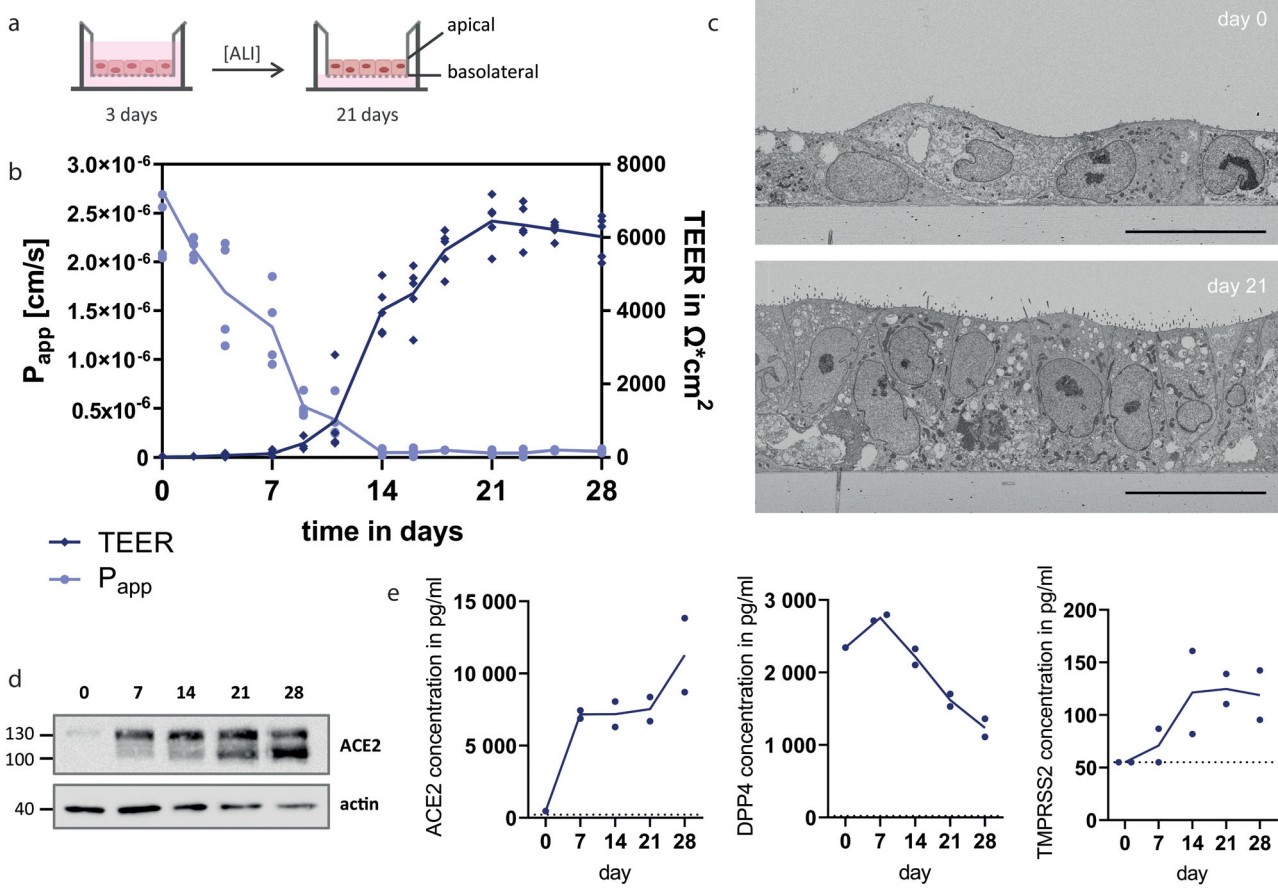

**Fig. 1 hAELVi cell air–liquid-interface cultures as model of the distal lung. a** Schematic representation of hAELVi air–liquid-interface (ALI) cultures.
**b** Measurement of transepithelial electrical resistance (TEER) and permeability of sodium fluorescein across the cellular monolayer of hAELVi cells growing under ALI up to 28 days. Experiments were performed for five (TEER) or four ($P_{app}$) independent experiments in technical duplicates, respectively.
**c** Electron microscopic analysis of hAELVi cells grown at the ALI for 0 or 21 days. Data are representative for two independent experiments. Bar: 20 μm.
**d** Western Blot analysis of ACE2 expression of hAELVi cells grown under ALI for 0, 7, 14, 21, or 28 days, respectively. Data are representative for three independent experiments. **e** Detection of ACE2, DPP4, and TMPRSS2 in cell lysates of hAELVi cells grown under ALI for 0, 7, 14, 21, or 28 days, respectively, using ELISA. Experiments were performed for two independent experiments in technical duplicates.

hAELVi cultures with viral titers peaking at 48 h p.i. at $5 \times 10^7$ pfu/ml for SARS-CoV-2, $2 \times 10^8$ pfu/ml for SARS-CoV and higher than $1 \times 10^9$ pfu/ml for seasonal IAV (Fig. 2b). TEER values of hAELVi ALI cultures decreased upon SARS-CoV-2 infection to some extent compared to non-infected cultures, suggesting that barrier properties of the epithelial cell layer were impaired but not destroyed by virus infection during the observation time (Supplemental Fig. 3). Peak titers of MERS-CoV were up to two orders of magnitude lower compared to SARS-CoV-2. Confocal laser-scanning microscopy confirmed the infection by SARS-CoV-2 or IAV in single, separated cells at 16 h p.i.. Throughout the course of infection, the size of infected cell clusters increased, indicating ongoing viral cell-to-cell spread (Fig. 2c). Thin-section electron microscopy demonstrated the presence of large numbers of viral particles at the surface of (Fig. 2d) and within (Supplemental Fig. 4) single cells or small groups of cells intermingled between non-infected epithelial cells. Extracellular virus particles were linked by fine fibrous material (Fig. 2d), which was present at the surface of all cells (Supplemental Fig. 4c). All ultrastructural hallmarks of coronavirus replication, such as double-membrane vesicles, budding into membrane compartments and release at the cell surface, were visible in the infected cultures (Supplemental Fig. 4).

To assess the hAELVi model for early immune activation by the investigated viruses we analyzed samples taken from the basolateral compartment of infected cells for the presence of type I and III interferons (IFN), as well as selected cytokines and chemokines by ELISA. At early time points (16 h p.i.) we observed no significant release of most mediators with the exception of low levels of CXCL10 in SARS-CoV infected hAELVi cultures. Interestingly, at 48 h p.i. SARS-CoV-2 infection triggered secretion of IFNβ and λ1-3 at moderate levels, whereas this was more pronounced with IAV (Fig. 2d). SARS-CoV-2 and SARS-CoV induced distinct sets of proinflammatory cyto- and chemokines. Whereas SARS-CoV upregulated a larger set of immune factors, including CXCL1, CXCL5, IL-8, MIF, and CXCL10, SARS-CoV-2 triggered only modest secretion of MIF and CXCL10 from infected cells (Fig. 2e).

**Attenuated replication of SARS-CoV-2 Omicron variant.** Finally, we utilized the hAELVi model to compare propagation of patient isolates from the recently emerged SARS-CoV-2 Omicron BA.1 and BA.2 lineages to SARS-CoV-2 D614G and the Delta variant at two different doses of infection (Fig. 2a, b). The analysis showed similar replication kinetics for the D614G and Delta viruses with highest titers between 2.5 and $5.5 \times 10^7$ pfu/ml at 48 h (MOI = 0.3) and $1.9–2.3 \times 10^7$ pfu/ml at 72 h p.i. (MOI = 0.03), respectively. A slight decrease in viral titers in the late stage of infection with SARS-CoV-2 D614G was associated with a cytotoxic effect as demonstrated by elevated lactate

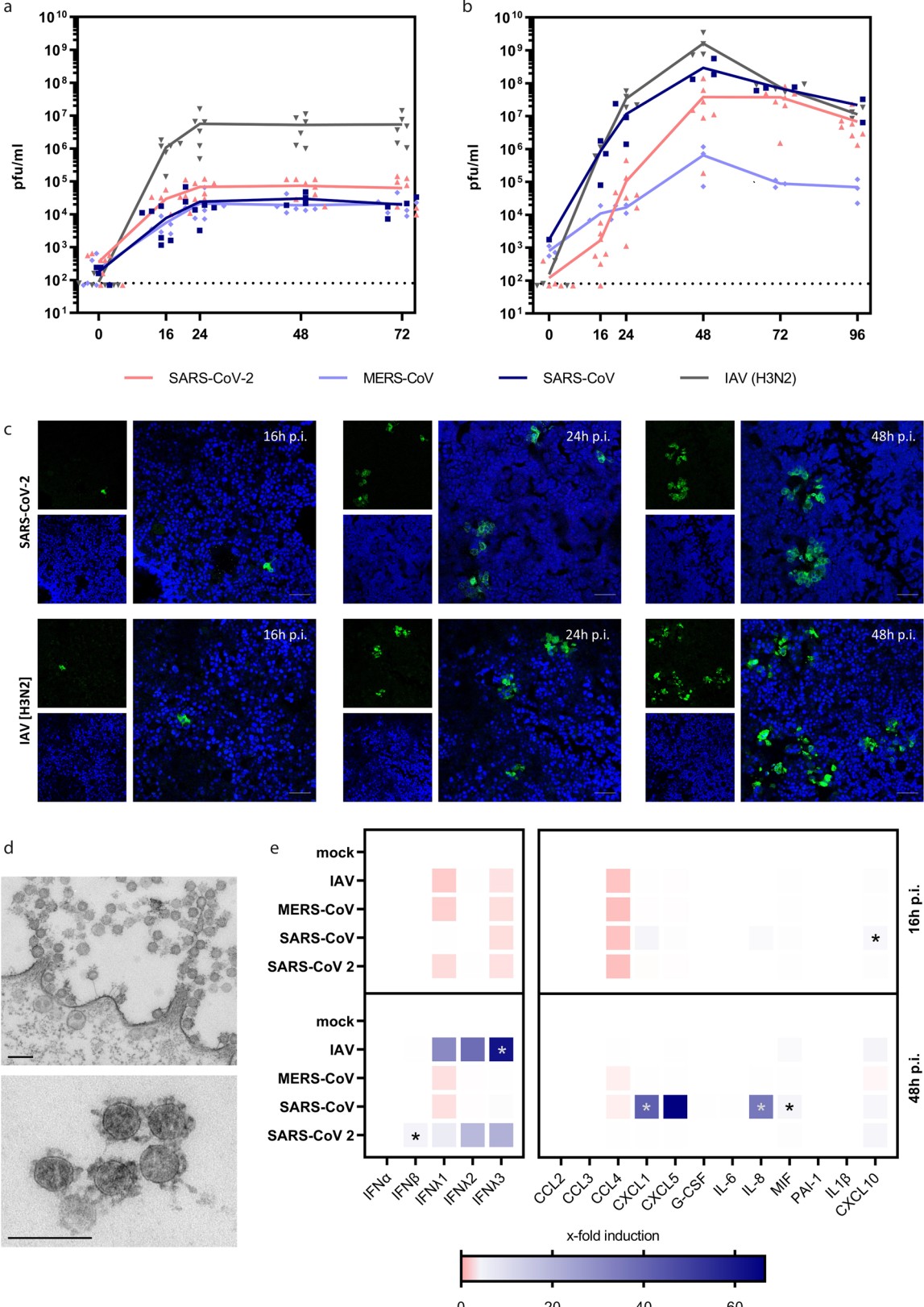

dehydrogenase (LDH) concentration measured in apical washes at 72 h p.i. (Supplemental Fig. 6). Interestingly, compared to Delta, replication of the Omicron BA.1 and BA.2 isolates was strongly reduced by about 2 and 3 orders of magnitude, respectively, at 48 h p.i. (Fig. 3a). This effect was similar following infection with a 10-fold decreased dose of infection, when BA.1 and BA.2 titers were about 3 orders of magnitude lower at 48 h p.i. compared to D614G or Delta (Fig. 3b). Interestingly, in a lentiviral pseudotype assay, the Omicron spike protein facilitated more efficient entry into human epithelial HT1080 cells over-expressing ACE2 than the spike of the Delta variant[15] (Supplemental Fig. 7). However, the Omicron spike was less active than

**Fig. 2 Infection of hAELVi cell air–liquid-interface cultures with highly pathogenic coronaviruses. a–e** Submerged hAELVi cells (**a**) or hAELVi cell grown under ALI for 21 days (**b–e**) were used for infection experiments with highly pathogenic coronaviruses. **a + b** Cells were infected with SARS-CoV-2 D614G, SARS-CoV, MERS-CoV, and IAV at MOI of 0.3 and further incubated under ALI conditions at 37 °C. Progeny viruses were collected at indicated time points and titrated using standard plaque assay on VeroE6 cells. For ALI cultures (**b**), washes from the apical compartment were performed to collect supernatants. Replication analysis was performed for $n = 3$ in technical duplicates. **c** Cells were infected with SARS-CoV-2 or IAV at MOI 1 and were processed for analysis by confocal laser-scanning fluorescence microscopy to detect the viral spike protein (SARS-CoV-2) (SARS-CoV-2 Spike Antibody, Sino Biological) or M2 protein (IAV) (IV A/M2 antibody, Thermo Fisher Scientific) at 16 h, 24 h, and 48 h p.i. (green channel). Nuclei were stained with DAPI (blue channel). Data are representative for two independent experiments. Bar: 50 μm. **d** Thin-section electron microscopy of a SARS-CoV-2-infected epithelial cell shows cluster of coronavirus particles at the cell surface (upper image) and a small cluster at higher magnification. Infection was performed with MOI of 3 and cells were fixed 24 h p.i.. Data are representative for two independent experiments. Bar: 200 nm. **e** Cells were infected with SARS-CoV-2, SARS-CoV, MERS-CoV and IAV at MOI of 1 followed by collection of the basolateral fluid at 16 h and 48 h p.i. for ELISA detection of type I and III IFN or the indicated cyto- and chemokines. Experiments were performed for three independent experiments in technical duplicates. Statistical analysis was done by using Kruskal–Wallis test, *$p < 0.05$.

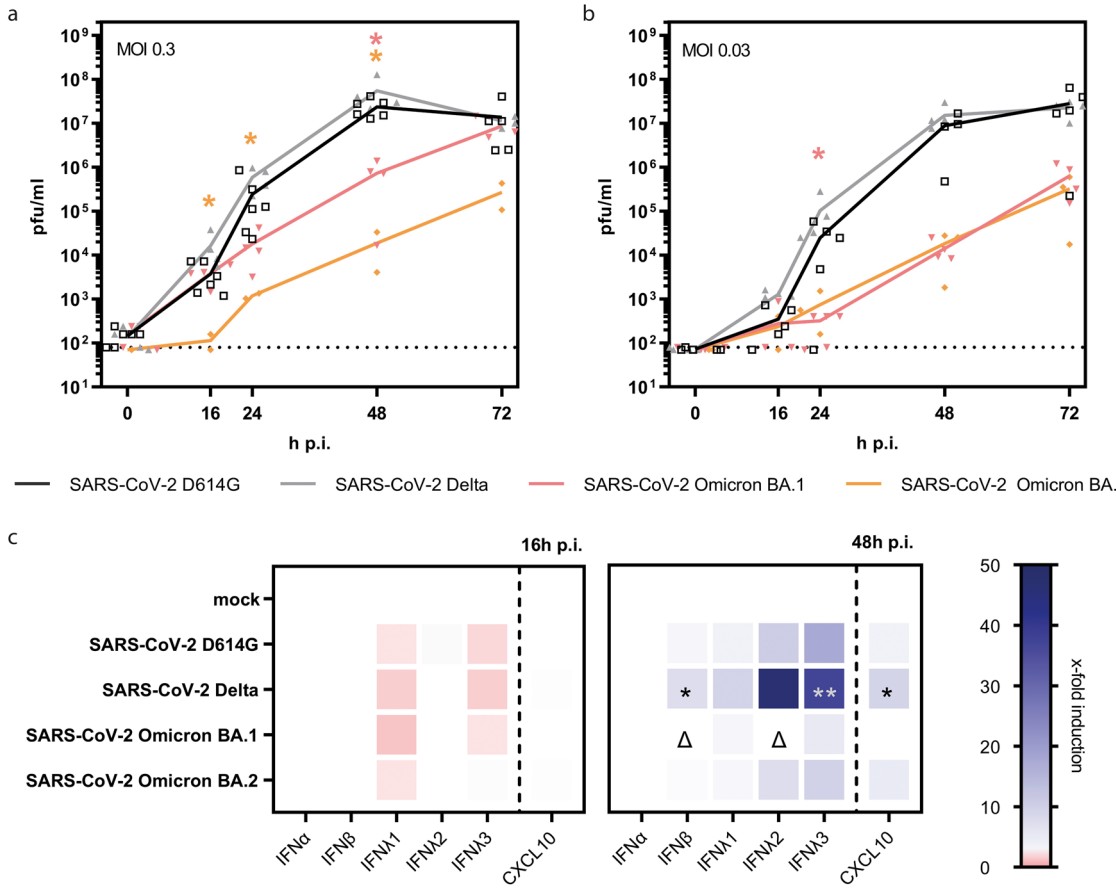

**Fig. 3 Infection of hAELVi cell air–liquid-interface cultures with SARS-CoV-2 VOC. a–c** hAELVi cells grown under ALI for 21 days were used for infection experiments with SARS-CoV VOC. Cells were infected with MOI 0.3 (**a**) or MOI 0.03 (**b**) or MOI 1 (**c**) with SARS-CoV-2 variants D641G, Delta, Omicron BA.1 or BA.2, respectively. **a + b** For growth curve analysis, apical washes were performed at indicated time points and titrated using standard plaque assay. Replication analysis was performed for $n = 2$ in duplicates. Statistical significance is displayed compared to SARS-CoV-2 Delta VOC infected cells. **c** Basolateral fluids of infected cells were collected at 16 h and 48 h p.i. for ELISA detection of type I and III IFN or the indicated cyto- and chemokines. Experiments were performed for three independent experiments in technical duplicates. Statistical significance compared to mock-infection is indicated by * and Δ compared to Delta infected samples. Statistical analysis was done by using Kruskal–Wallis test, *$p < 0.05$, **$p < 0.01$.

Delta when probed for entry into polarized hAELVi cells, which may at least partially explain attenuated replication of Omicron in these lung cell cultures. We finally evaluated the variants for immune activation by assaying selected mediators for which we had previously observed noticeable induction by SARS-CoV-2 (Fig. 1e). This analysis revealed little cytokine release at 16 h p.i., but strong induction of IFNβ, IFNλ1-3, and CXCL10 for Delta at 48 h p.i.. Cell activation by Omicron isolates was notably decreased compared to D614G and Delta (Fig. 3c). Collectively, these findings demonstrate for the Omicron variant an attenuated

growth phenotype as well as reduced cytokine activation in a human ALI infection model of the distal lung in comparison to previous SARS-CoV-2 including Delta.

## Discussion

The alveolar space of the human lung can be heavily affected in severe COVID-19 or influenza, but analyses of viral replication and pathophysiological processes are limited by the inability to systematically sample infected cells of the alveolar epithelium in

severely diseased patients *intra vitam*. In our study we present several pieces of evidence that cultures of polarized hAELVI cells grown under ALI can model productive SARS-CoV-2 infections in the lower respiratory tract under controlled experimental conditions. This includes virion morphogenesis and release, cell-to-cell spread, accumulation of high viral titers, as well as moderate induction of antiviral type I and III IFNs.

Besides active viral replication it has been shown that induction of regulatory cytokines plays a critical role in the pathogenesis of COVID-19[16]. Our study showed that hAELVi cultures maintain the capacity to react towards a viral stimulus. Upregulation of various proinflammatory mediators in this model is in line with previous analyses, suggesting significant activation of cytokines and chemokines, including IL-8, CXCL1, CXCL5, and CXCL10 in human lung explants infected with SARS-CoV ex vivo, whereas SARS-CoV-2 induced only a smaller subset of these mediators[17]. Moreover, virus-infected hAELVi cultures did upregulate type I and type III IFN suggesting their suitability to scrutinize in future studies the role of these antiviral and antiproliferative cytokines to tissue damage observed in the lower respiratory tract of patients with severe to critical COVID-19[16]. The possibility to establish co-cultures with macrophages-like cells raises the opportunity to extend the hAELVi infection model for future investigations of the cross-talk between epithelial and myeloid cells[18].

In the present study, we compared propagation of recently emerged Omicron BA.1 and BA.2 isolates with the earlier D614G and Delta variants in the polarized human hAELVi lung cell cultures infected as a proxy for viral fitness in the distal lung. Our analysis demonstrated an attenuated growth phenotype of both Omicron isolates in comparison to the earlier variants as viral titers increased only in a delayed or reduced manner. As one potential contributor to viral attenuation we found that the Omicron spike was less efficient in mediating cell entry into hAELVi cultures in a pseudotype virus particle assay compared to the Delta spike. We suggest that the reduced uptake of Omicron spike in hAELVi cells is linked to its recently described preference to enter cells via the endosomal route of entry, which is independent of TMPRSS2-assisted cell entry employed by earlier variants including Delta[6,19]. However, further investigations are required to evaluate this hypothesis. Notably, reduced fitness of Omicron in the hAELVi ALI model mirrors early conclusions on reduced virus titers in the lung and disease severity caused by Omicron infections in hamster and mouse models[20,21], and recapitulates and extends recent results obtained in explanted human lung tissue[22]. In contrast to findings of reduced replication for Omicron viruses in the lower respiratory tract compared to other VOC, recent studies showed an increased replication capacity in primary human nasal airway epithelial cells, suggesting that the enhanced spread of the Omicron lineage is due to an increased fitness of the variant in the upper respiratory tract[6,19]. Overall, our experimental findings are well in line with epidemiological observations showing a reduced likelihood for hospitalization of COVID patients infected with SARS-CoV-2 Omicron compared to Delta[7,8,22]. Collectively, we suggest that hAELVi cultures are an experimental human infection model that is well suited for fast phenotypic evaluation of emerging viral variants and to inform Public health strategies in a timely manner.

## Methods

**Cell culture**. hAELVI cells were obtained from inSCREENex (Cat INS-CI-1015). For cultivation, cell culture flasks were coated prior to application of cells using huAEC Coating solution (inSCREENex, INS-SU-1018-100 ml). hAELVi cells were cultivated in huAEC Medium (inSCREENex, INS-ME-1013-500 ml). Prior to seeding of hAELVi cells on transwell filters (ThinCert, pore size 0.4 µm, Greiner), inserts were coated by adding huAEC coating solution according to the manufactures protocol. Approximately $2.3 \times 10^5$ (12-well: Figs. 1b–e, 2, 3c, Supplemental

Figs. 3 and 8) or $0.7 \times 10^5$ cells (24-well: Fig. 3a, b and Supplemental Fig. 6) were seeded into the apical chamber of the filter insert and initially incubated at 37 °C and 5% $CO_2$ under liquid–liquid-conditions for 3 days following further cultivation under ALI up to 28 days. Based on the analyses of barrier integrity, morphological analysis and expression of cellular host factors, the polarized hAELVi cells were used for infection experiments after an incubation period of at least 21 days under ALI. Vero E6 cells, HEK 293T cells and hACE2 expressing HT1080-ACE2 cells[23] were propagated in Dulbecco's modified Eagle medium (DMEM) containing 10% fetal bovine serum (FBS) supplemented with 2 mM L-glutamine, 100 U/ml penicillin, 100 µg/ml streptomycin, 1x non-essential amino acids, and 1 mM sodium pyruvate. MDCKII cells were cultivated in minimum essential medium (MEM) containing 10% fetal bovine serum (FBS), 2 mM L-glutamine, 100 U/ml penicillin and 100 µg/ml streptomycin. All cells were incubated at 37 °C with 5% $CO_2$ in a humidified atmosphere.

**Infection of hAELVi cells and virus titration on Vero E6 cells**. Cells were infected with SARS-CoV-2 D614G (ENA project PRJEB55524; accession ID ERS12788651), SARS-CoV-2 Delta (ENA project PRJEB50616; sequence ID IMSSC2-206-2021-00148), SARS-CoV-2 Omicron BA.1 (ENA project PRJEB55524; accession ID ERS12788650), SARS-CoV-2 Omicron BA.2 (ENA project PRJEB55524; accession ID ERS12788649), SARS-CoV Frankfurt-1 (Genbank accession number FJ429166.1), MERS-CoV EMC/2012 (Genbank: JX869059) or Influenza A/Panama/2007/1999 virus (Genbank: DQ487333-DQ487340). For infection of submerged cultures, cells were washed with PBS and incubated with D-PBS/0.3% BA containing the appropriate amount of virus for 1 h at 37 °C. Subsequently, cells were washed with PBS and fresh medium was added. For infection of ALI cultures, cells were washed with D-PBS once and inoculated with virus diluted in D-PBS/0.3% BA in the apical chamber. Following incubation for 1 h at 37 °C, cells were washed apically with D-PBS and fresh medium was added to the basolateral compartment of the filter insert. For replication analysis, 10% of the supernatant (submerged) or culture medium in the basolateral compartment (ALI), respectively, was harvested at indicated time points and refilled with fresh culture medium. To collect apical samples 50 µl (24-well) or 100 µl D-PBS (12-well), respectively, were used to perform apical washes at 37 °C for 30 min. Supernatants were stored at −80 °C until titration by standard Plaque Assay on Vero E6 cells (CoV) or MDCKII cells (influenza A virus) using Avicel overlay was performed to quantify infectious virus particles.

**Ethical declaration**. SARS-CoV-2 viruses were isolated from naso- or oropharyngeal swabs on VeroE6 or Caco-2 cells in the context of routine diagnostics by RKI units FG17 and ZBS1, which were approved by the ethics committees of Charité-Universitätsmedizin Berlin Ethical Board (Reference EA2/126/11) and the Berlin Medical Association (Eth-40/20).

**Preparation of cell lysates and immunoblot analysis**. Cells were washed twice with ice-cold D-PBS before lysis buffer (10 mM Tris/HCl (pH 7,5), 150 mM NaCl, 0.5 mM EDTA, 1% NP-40 including protease inhibitor) was added to the apical chamber of the filter insert and incubated for at least 30 min at 4 °C. Cells lysates were centrifuged at $15,000 \times g$ and 4 °C for 10 min and supernatants were stored at −20 °C until further processing. Fifty micrograms of each sample was separated by reducing SDS-PAGE under denaturing conditions and transferred onto nitrocellulose membrane by semi-dry western blotting. Detection of ACE2 was performed by incubation with anti-hACE2 antibody (R&D Systems (AF933); 1:500) and suitable secondary antibody coupled to horseradish-peroxidase (HRP) (1:10,000; Agilent Technologies, Santa Clara, USA). Equal loading of samples was controlled with immunostaining of actin. SuperSignal™WestDura Extended Duration Substrate was added to the membrane and the resulting chemiluminescence was detected using an Advanced Fluorescence Imager (Intas). Uncropped Western Blot membranes are depicted in Supplemental Fig. 9.

**Cytokine ELISA**. Samples taken from the basolateral compartment of infected cells were collected at indicated time points and stored at −80 °C until further processing. If necessary, samples were diluted in the corresponding culture medium prior to ELISA measurement. Samples were analyzed using the R&D DuoSet ELISA Kits DY9345 (IFNα2), DY814 (IFNβ), DY7246 (IFNλ1), DY1587 (IFNλ2), DY5259 (IFNλ3), DY279 (CCL2), DY270 (CCL3), DY271 (CCL4), DY275 (CXCL1), DY254 (CXCL5), DY214 (G-CSF), DY206 (IL-6), DY208 (IL-8), DY289 (MIF), DY201 (IL-1β), DY266 (CXCL10) according to the manufacturer's instructions. For collective heat map presentation of all analyzed cytokines, mean values of x-fold inductions relative to mock-infection of three independent experiments were calculated for each cytokine. Individual concentrations of all cytokines are shown in Supplementary Fig. 5 (highly pathogenic coronaviruses) or 8 (SARS-CoV-2 VOC), respectively.

For quantification of host factor expression cell lysates were prepared as previously described, total protein amount was measured using -BCA-Kit-according to the manufacturer's instructions and 100 µg of protein lysate was analyzed using the R&D DuoSet ELISA Kits DY933-05 (ACE2), DY1180 (DPP4) and the commercially available ELISA kit ABIN6960140 (TMPRSS2) according to the corresponding manufacturers protocol.

**Detection of viral antigen by laser-scanning confocal fluorescence microscopy**. Cells were washed once with D-PBS, fixed with 3.7% formaldehyde in D-PBS for 15 min at RT and further incubated for 10 min at RT with 10 mM ammonium chloride/D-PBS. Subsequently, cells were permeabilized for 7 min with 0.5% TritonX-100/D-PBS at RT, followed by blocking of cells with 3% BSA/D-PBS for at least 1 h at RT. Between each of these steps, the samples were washed twice with D-PBS.

For staining of specific proteins, filter inserts were stamped out with 6 mm biopsy punches and incubated with primary antibodies diluted in 3% BSA/D-PBS at 4 °C overnight. After washing three times with D-PBS for at least 5 min, cells were incubated with secondary fluorescence conjugated antibodies for 1 h at RT in the dark. Cells were subsequently washed twice with D-PBS and nucleus stained by incubating cells with DAPI for 10 min. Fluorescence laser-scanning microscopy was performed using a Zeiss LSM 780 confocal microscope with corresponding ZEN software and a Plan-Achromate 20x (NA 0.8) objective.

**Thin-section electron microscopy (EM)**. hAELVi cells were fixed on their filter substrate with a mixture of 1% formaldehyde and 2.5% glutaraldehyde in HEPES (0.05 M, pH 7.4) for at least 2 h at RT. Filters were punched out with a skin punch and were post-fixed with osmium tetroxide, tannic acid, uranyl acetate and embedded in epon[24]. Ultrathin sections were collected on naked grids and examined with a transmission electron microscope (Tecnai Spirit, Thermo Fisher Scientific Inc.) operated at 120 kV acceleration voltage. Images were recorded with a CMOS camera (Phurona, EMSIS). Overview images of the epithelium were recorded from carbon coated resin block-faces after sectioning with a diamond knife using a scanning electron microscope (Teneo VS, Thermo Fisher Scientific Inc.) and the T1 detector at 2 kV.

**Determination of transepithelial electrical resistance (TEER) and transepithelial transport of fluorescein in hAELVi cell monolayers**. To determine TEER values, the apical chamber of cultured hAELVi cells was refilled to LCC levels with D-PBS and the MillicellR ERS (electrical resistance system) volt was used for measurement. To obtain $\Omega \ast cm^2$ the surface area of the transwell insert was multiplied with the measured TEER values. To assess the transport of sodium fluorescein across the cellular monolayer, fresh culture medium was added to the basolateral acceptor chamber and culture medium containing 1 mg/ml sodium fluorescein was added into the apical chamber of the insert. Every 30 min for 3 h one-fifth of the basolateral culture medium was taken and directly replaced with fresh huAEC medium. Optical density of samples was determined at 486 nm including a sodium fluorescein standard curve (0, 5, 10, 15, 20, 30, 40, 50 μg/ml).

**Lactate dehydrogenase (LDH) assay**. Cells were washed with D-PBS once and apically inoculated with MOI 0.3 or 0,03 of SARS-CoV-2 D614G, SARS-CoV-2 Omicron BA.1 or SARS-CoV-2 Omicron BA.2 diluted in D-PBS/0.3% BA. Following incubation for 1 h at 37 °C, cells were washed apically with D-PBS and fresh medium was added to the basolateral compartment of the filter insert. To collect samples 100 μl D-PBS were used to perform apical washes at 37 °C for 30 min. which were stored at −80 °C. To quantify LDH release, samples were diluted (1:500) and LDH-Glo™ Cytotoxicity Assay (Promega J2381) was performed according to the manufacturer's instructions.

**HIV-based pseudotype virus particles**. The pseudo particles were generated using a CMV-driven human codon-optimized expression plasmid of C-terminal truncated (Δ19 amino acids) SARS-CoV-2 spike variant (pSARS-CoV-2_variant_Δ19aa, variants: Wuhan [unmodified sequence]; Delta [B.1.617.2]; BA.2) and the replication-defective lentiviral HIV-1 backbone with depleted Env and inserted NanoLuciferase (pHIV-1NL4-3ΔEnv-NanoLuc) as described[23]. In brief, 5 × 10⁶ HEK 293T cells were seeded in 10 cm dishes and transfected the next day with 2.5 μg pSARS-CoV-2_variant_Δ19aa and 7.5 μg pHIV-1NL4-3ΔEnv-NanoLuc mixed with 25 μg PEI in serum-free 1x DMEM. After 8 h, the cells were washed twice with 1x DMEM and incubated in 10 ml 1x DMEM. At 48 h post-transfection, the supernatant was harvested and centrifuged (5 min, 300 × g) to remove cell debris, filtrated with a cell strainer (0.22 μm pore size), aliquoted in 100 μl and stored at −80 °C.

In preparation for the infectivity assay, pseudo particles were lysed in 1x RNA lysis buffer (10 mM Tris, 150 mM NaCl, 0.25% IPEGAL, pH 7.5)[25,26] to quantify concentration as virus-like particles per ml (VLP/ml) by RT-qPCR (RevertAid Reverse Transcriptase, Thermo Scientific™; Expand™ High FidelityPLUS PCR-System, Roche) using a DNA standard and a TaqMan probe that binds to the gag gene of the HIV-1 genome as described[27]. The cells were transduced with pseudo particles depending on the cell type and scale. After washing with D-PBS (basolateral and apical for cells grown on filter inserts), hAELVi ALI cultures were infected with 150 μl of pseudo particles on the apical side and HT1080-ACE2 cells (96-well plate) with 50 μl. After 1 h incubation at 37 °C, cells were washed with D-PBS twice and incubated with 1x DMEM. At 24 h p.i., HT1080-ACE2 cells were washed twice with D-PBS and lysed in 50 μl 1x Luciferase Cell Culture Lysis Reagent (Promega). hAELVi ALI cultures were washed once basolateral with D-PBS at 48 h p.i. and were subsequently directly incubated with 100 μl 1x Luciferase Cell Culture Lysis Reagent apically. Cells were lysed at RT for 30 min, and half the volume of the lysates were transferred to a white solid flat-bottom 96-well plate.

The NanoLuciferase activity was measured using Nano-Glo® Luciferase Assay System (Promega). The lysates were mixed 1:1 with fresh-prepared NanoGlo buffer containing substrate (1:51) and incubated for 3 min at RT. The bioluminescence was measured with the luminometer TriStar LB 941 (Berthold Technologies) with an integration time of 0.1 s to obtain relative light units (RLU) as a read-out for pseudotyped entry and hence infectivity.

**Statistics and reproducibility**. All statistical analyses were done by using non-paired, non-parametric Kruskal–Wallis test (*$p < 0.05$) or two-sided, unpaired Student's t-test (*$p < 0.05$) as indicated and performed using GraphPad Prism Software Version 9.1.0. Results were presented as mean ± standard error (SEM). For heat map presentation, mean values of x-fold inductions relative to mock-infection of three independent experiments were calculated for each cytokine.

**Reporting summary**. Further information on research design is available in the Nature Research Reporting Summary linked to this article.

## Data availability
The datasets generated and analyzed during the current study are available from the corresponding author on reasonable request.

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

## Acknowledgements

We thank Gudrun Heins for excellent technical assistance. We thank Paul Bieniasz (The Rockefeller University, New York) for the pHIV-1NL43ΔEnv-NanoLuc construct, the SARS-CoV-2-SΔ19 (Wuhan) plasmid and the HT1080/ACE2 cell line. This work was supported by projects RAPIDII (01KI2006F) and NUM-COVID-19 Organo-Strat funded by the Federal Ministry of Education and Research (BMBF), as well as by the TransRegio 84 of the German Research Foundation (project B2) (T.W.).

## Author contributions

C.M., J.S., J.-M.G., and D.B. planned and conducted the experiments and analyzed the data. G.H. and M.L. performed electron microscopic analysis. C.M., J.S., D.-Y.O., M.L., A.N., R.D., and T.W. contributed to the interpretations and conclusions presented. C.M., J.S., M.L., and T.W. wrote the manuscript. All authors participated in editing the manuscript.

## Funding

## Competing interests

The authors declare no competing interests.
