## [Peer Review File · Communications Biology]

Reviewers' comments:

Reviewer #1 (Remarks to the Author):

Mache et al use a polarized Alveolar Epithelial model to examine the replication dynamics of important respiratory pathogens such as SARS-CoV-1, SARS-CoV-2, MERS-CoV and influenza. They demonstrate that these cultures are permissive to viral infection, and they profile the immune activation induced by these pathogens. Finally, they demonstrate reduced replication of the SARS-CoV-2 Omicron VOC in this model of viral infection.

Major Comments:

Overall, the paper is well presented and written to a high standard and demonstrate the utility of hAELVi cultures in the study of respiratory pathogens. In figure 2A the comparison of SARS-CoV-2,1 and MERS with IAV is an interesting experiment and highlights the varied tropism of these viruses. I wonder have the authors considered assessing the specific infectivity of virus particles produced from these cultures, compared to virus produced from submerged cell culture models? Why was the cell to cell spread data presented in figure 2B only performed for SARS-CoV-2? I think it would be important to perform this analysis with the other viruses presented in figure 1A if the authors wish to demonstrate the utility of hAELVi cells as a model for respiratory pathogens.

Investigation of the immune response between the various CoV and IAV presented in figure 2D-E is again an interesting data set? Why was 48h selected as the time point for sampling the IFNs and cytokines? Looking at the production of these various markers over time would provide a more dynamic and informative data set allowing a more accurate comparison of the cellular response to these various viral infections.

In Figure 2E the reduction in PFU/ml at 72h in the D614G/delta infections, is this due to cytotoxicity? Do the authors have any viability data for these experiments? Given the accumulating evidence that omicron has an altered cell entry mechanism and reduced syncytia formation (see: bioRxiv, (2022), 2021.12.31.474653 and medRxiv, (2022), 2022.01.03.21268111) have the authors considered studying initial establishment of omicron in these cultures and in particular cell:cell spread? This could be of particular interest to model given the reduced replication kinetics of this variant. While perhaps beyond the scope of this paper, it would be very informative to compare replication of this model to nasal ALI cultures particularly in the context of this VOC.

Minor comments

Size markers on WB in figure 1C are missing

Line 98: Should be 'form' not 'from

Line 141-144: sentence is confusingly worded

Reviewer #2 (Remarks to the Author):

The authors of manuscript characterize polarized human Alveolar Epithelial cells that have been immortalized by lentivirus to be used as a model for virus infection of the lower human respiratory tract.

Under air-liquid interface culturing, SARS-CoV and SARS-CoV-2 (an early isolate) replicate efficiently given the expression of SARS host factors, ACE2 and TMPRSS2 on this particular cell lines. Influenza virus also replicates robustly while MERS-CoV-2 is attenuated in this cell line. The attenuation of MERS is correlated to the lower expression of its cellular receptor DPP4 in this cell line.

The authors also examined the induction of typical cytokines and chemokines induced by infection.

Finally, the authors included the SARS-CoV-2 variant Omicron into growth kinetics studies. The Omicron variant lagged behind in replication but replicated to the same titer by 72 hours. But at a 10-fold lower input of starting virus, Omicron replication was more attenuated compared to the early isolate.

From their studies, the authors suggest that the immortalized human Alveolar Epithelial cells can be used for "fast phenotypic evaluation of emerging viral variants and to inform Public health strategies in a timely manner."

To make this conclusion, the authors need additional data comparing different isolates including Omicron. There seems to be a minimal data set to make this conclusion. The authors only provide one set of comparison data for Omicron – virus replication.

How does Omicron compare to other isolates/variants in terms of cytokine/chemokine induction?

The authors started with data on the characterization of this cell line describing transepithelial resistance; how does this change in terms of virus infection between different viruses/isolates/variants? How do the overall protein levels or phosphorylation states of proteins related to tight junctions change during infection? by different variants?

The fact that this cell line has already been described and generally characterized plus the limited virological studies, weakens the manuscript and the conclusion drawn by the authors.

Minor

The model is not novel; was described by Kuehn A et. al ALTEX 33: 251-60 2016 as cited by the authors of this manuscript.

With BA.1 and BA.2, it is important for the authors to indicate in the text which Omicron variant is presented in this manuscript.

The authors cite Peacock et. al but don't discuss the fact that the Peacock manuscript demonstrated better Omicron replication in nasal derived cells. This point between the two different cell types and replication ability may demonstrate differences between transmission and pathogenicity of variants and should be mentioned in some degree.

POINT-TO-POINT RESPONSES:

Reviewer 1:

Mache et al use a polarized Alveolar Epithelial model to examine the replication dynamics of important respiratory pathogens such as SARS-CoV-1, SARS-CoV-2, MERS-CoV and influenza. They demonstrate that these cultures are permissive to viral infection, and they profile the immune activation induced by these pathogens. Finally, they demonstrate reduced replication of the SARS-CoV-2 Omicron VOC in this model of viral infection.

Major Comments:

Overall, the paper is well presented and written to a high standard and demonstrate the utility of hAELVi cultures in the study of respiratory pathogens. In figure 2A the comparison of SARS-CoV-2,1 and MERS with IAV is an interesting experiment and highlights the varied tropism of these viruses. I wonder have the authors considered assessing the specific infectivity of virus particles produced from these cultures, compared to virus produced from submerged cell culture models?

Thank you for these comments. In response, we provide, in the new figure 2a, additional replication curves of the same set of viruses (IAV, SARS-CoV, MERS-CoV, SARS-CoV-2) on submerged hAELVi cultures, to enable a direct comparison with viral replication in polarized ALI cultures (now labelled as figure 2b). The data demonstrates a remarkable increase of viral titers in polarized ALI cultures compared to the usual submerged cultivation by almost three orders of magnitude for SARS-CoV-2, SARS-CoV and IAV. We suggest that elevated replication of SARS-CoV and SARS-CoV-2 may be a result of enhanced host factor expression, as levels of ACE2 and TMPRSS2 increase during the polarization process, as shown in panel 1e. The corresponding results are described in the result (line 114-120) and the methods sections (line 233-235) and further necessitated changes in legend of figure 2 (line 470-474). Assessment of the specific infectivity of viruses grown in the two culture settings would be interesting but should also consider the infectivity of viruses produced by other infection models and, thus, remains to be determined in future work.

Why was the cell to cell spread data presented in figure 2B only performed for SARS-CoV-2? I think it would be important to perform this analysis with the other viruses presented in figure 1A if the authors wish to demonstrate the utility of hAELVi cells as a model for respiratory pathogens.

We performed additional experiments to study also the viral spread of influenza A virus (IAV) in hAELVi cultures and extended the observation period to an earlier time point of infection (16 h p.i.). The results are presented in the new figure 2c. Here, we were able to demonstrate ongoing cell-to-cell spread of the infection for both viruses as indicated by expansions of cell clusters that stain positive for viral antigen with increasing time post infection. The results of the additional investigations are described in the results (lines 123-126) and in legend of figure 2 (line 475-478).

Investigation of the immune response between the various CoV and IAV presented in figure 2D-E is again an interesting data set? Why was 48h selected as the time point for sampling the IFNs and cytokines? Looking at the production of these various markers over time would provide a more dynamic and informative data set allowing a more accurate comparison of the cellular response to these various viral infections.

We agree that a kinetic component in the investigation of virus-induced inflammatory mediators provides additional valuable insights into the dynamics of immune activation. With the new figure 2e we complemented our data on the secretion of type I and III IFNs, as well as of selected cytokines, for the time-point representing peak virus replication (48h) with data for the early phase of infection (16h). At 16h p.i., we observed only low levels of immune induction with slight upregulation of CXCL10 upon

infection with SARS-CoV. Quantification of immune factors released by 48h p.i. revealed a more pronounced induction pattern in IAV and SARS-CoV-2 infected cells. The new results are described in the results section lines 133-141 and the legend of figure 2 (lines 483-485). We accordingly updated findings presented in supplementary figure 5, which details absolute cytokine levels.

In Figure 2E the reduction in PFU/ml at 72h in the D614G/delta infections, is this due to cytotoxicity? Do the authors have any viability data for these experiments?

Thank you very much for bringing this up. To investigate whether reduction of viral titers at 72h p.i. was a result of cytotoxicity or not, we determined Lactate dehydrogenase (LDH) release in available supernatants from hAELVi cultures infected with D614G, Omicron BA.1 or BA.2. We only observed an increase in LDH concentration in culture supernatants of cells infected with D614G MOI 0.3 at 72h p.i. which suggests a cytopathic effect (CPE) during late stages of infection. No significant LDH release was measured in Omicron-infected cultures that produced less progeny viruses than D614G- or Delta-infected cultures, suggesting that there is a threshold for CPE manifestation in the cultures that was not surpassed by BA.1 and BA.2. At this point we do not speculate about the underlying reason(s), and therefore just mention this additional finding in the text on lines 148-151, which is displayed in the new supplemental figure 6 in the amended manuscript. Further, we updated the method section in lines 317-324.

Given the accumulating evidence that omicron has an altered cell entry mechanism and reduced syncytia formation (see: bioRxiv, (2022), 2021.12.31.474653 and medRxiv, (2022), 2022.01.03.21268111) have the authors considered studying initial establishment of omicron in these cultures and in particular cell:cell spread? This could be of particular interest to model given the reduced replication kinetics of this variant.

This comment addresses an important point regarding the possible mechanistic reasons for the reduced ability of Omicron viruses to propagate in the lower respiratory tract of infected hamster and mouse models compared to earlier variants (e.g. Halfmann et al., 2022; Uraki et al., 2022). The literature mentioned by the referee suggests that the spikes of Omicron BA.1 and BA.2 favor an endosomal entry into host cells, which may be associated with a reduced usage of, or dependency on cleavage by TMPRSS-2. However, in our view it is far from being clear why these properties attenuate viral replication in the lower respiratory tract (LRT) of rodents and, possibly, humans. To follow up on these findings for the hAELVi cultures, we compared spike-mediated entry into polarized hAELVi cultures and an ACE-2 overexpressing human epithelial cell line (HT1080-ACE2) as a reference by using a pseudovirus approach (Schmidt et al., 2020). Interestingly, the competence of spike-mediated entry into HT1080-ACE-2 epithelial cells followed in essence recent findings by Willett et al. in human epithelial HEK293 cells (Willett et al., 2022) with Omicron BA.2 facilitating more efficient entry than Delta. In contrast, in hAELVi cultures we noted a reduced infectivity by Omicron spike compared to the Delta. This new dataset demonstrates a reduced capacity of Omicron BA.2 spike to mediate entry in alveolar hAELVi ALI cultures, which is likely to contribute to the attenuated growth of the Omicron viruses in these lung cells. We agree, it is important to gain further mechanistic insights into this observation such as the preferred mode of entry (e.g. endosomal, plasma membrane) and the specific host proteases involved (e.g. TMPRSS2, cathepsins). However, this is beyond the scope of the current study and will be addressed in ongoing and future work.

The new findings on cell entry are displayed in the new supplementary figure 7 and have been inserted in the results (lines 155-159), discussion (lines 190-194), methods (lines 326-353), and the acknowledgment sections (lines 433-435), and required addition of references 15, 19, 23 and 25-27.

While perhaps beyond the scope of this paper, it would be very informative to compare replication of this model to nasal ALI cultures particularly in the context of this VOC.

We share the reviewer's opinion that comparing ALI cultures from various origins is of high importance for evaluation of VOCs. Studies by Willet et al. and Peacock et al., which we mention in the discussion of the revised manuscript, demonstrate a replicative advantage of Omicron VOC in human nasal epithelial cell, which is accompanied by reduced replication in bronchial Calu-3 cells. However, such additional comparative analyses with hAELVi cultures will be subject of future investigations.

Minor comments

Size markers on WB in figure 1C are missing

Thank you very much for pointing this out. We added size markers in figure 1d.

Line 98: Should be 'form' not 'from

Thank you very much for bringing this up. We corrected this typo in line 101 of the revised manuscript

Line 141-144: sentence is confusingly worded

We clarified this sentence in the revised manuscript on lines 139-141.

Reviewer 2:

The authors of manuscript characterize polarized human Alveolar Epithelial cells that have been immortalized by lentivirus to be used as a model for virus infection of the lower human respiratory tract.

Under air-liquid interface culturing, SARS-CoV and SARS-CoV-2 (an early isolate) replicate efficiently given the expression of SARS host factors, ACE2 and TMPRSS2 on this particular cell lines. Influenza virus also replications robustly while MERS-CoV-2 is attenuated in this cell line. The attenuation of MERS is correlated to the lower expression of its cellular receptor DPP4 in this cell line.

The authors also examined the induction of typical cytokines and chemokines induced by infection. Finally, the authors included the SARS-CoV-2 variant Omicron into growth kinetics studies. The Omicron variant lagged behind in replication but replicated to the same titer by 72 hours. But at a 10-fold lower input of starting virus, Omicron replication was more attenuated compared to the early isolate.

From their studies, the authors suggest that the immortalized human Alveolar Epithelial cells can be used for "fast phenotypic evaluation of emerging viral variants and to inform Public health strategies in a timely manner."

To make this conclusion, the authors need additional data comparing different isolates including Omicron. There seems to be a minimal data set to make this conclusion. The authors only provide one set of comparison data for Omicron – virus replication.

We agree that the extension of the growth curve analyses to other VOCs of SARS-CoV-2 will support the suitability of the cell culture model. Therefore, we have included an additional Omicron isolate into our comparative analyses, now from the BA.2 lineage, which confirmed previous conclusions. Hence, the amended manuscript now presents cellular activation parameters and growth curve analyses of SARS-CoV-2 VOCs Delta, Omicron BA.1 as well as BA.2 compared to D614G virus across two different doses of infection (MOI of 0.3 and 0.03). The additional findings fully support our conclusion of an attenuated phenotype of Omicron in the hAELVi cultures as viral titers of BA.2 were reduced by 3 orders of magnitude at both MOI in comparison to Delta at 48 h p.i.. Moreover, an assessment of the release of type I and III IFN, as well as of CXCL10 from hAELVi cultures at 16 and 48 h p.i., identified

similar patterns of poor mediator induction for both BA.1 and BA.2 (see also answer to the following question).

The additional dataset on SARS-CoV-2 VOC replication is depicted in the new figure 3 in panels a and b. The results are described on lines 144-154 in the results, lines 228-231 in the methods section and necessitated further changes in lines 31, 54-55, 186-190 and 493.

How does Omicron compare to other isolates/variants in terms of cytokine/chemokine induction?

Analysis on immune activation of SARS-CoV-2 VOCs provides valuable information regarding evaluation of different viral variants. For the analyses of Omicron BA.1 and BA.2, and Delta VOCs we quantified selected IFNs and cytokines that were found upregulated upon infection with SARS-CoV-2 D614G (shown in panel 2e) in supernatants of hAELVi ALI cultures 16h and 48h p.i. by ELISA. At 16h p.i., we determined low levels of IFN λ 1 and IFN λ -3 for the Omicron and Delta variants. Interestingly, we observed little cytokine release at 16 h p.i., but a significantly stronger induction of IFN β , IFN λ 1-3 and CXCL10 levels by Delta infection at 48h p.i. compared to the two Omicron isolates. These differences may be ascribed either to a generally lower stimulation of cytokines by Omicron due to poorer replication or to the possibility that Omicron viruses are particularly well capable to control antiviral innate responses in the infected cell. This additional dataset on immune activation of SARS-CoV-2 variants is depicted in the new figure 3 panel c and necessitated addition of lines 162-163 in the results and additional changes in lines 496-498.

The authors started with data on the characterization of this cell line describing transepithelial resistance; how does this change in terms of virus infection between different viruses/isolates/variants? How do the overall protein levels or phosphorylation states of proteins related to tight junctions change during infection? by different variants?

In the original manuscript we showed that hAELVi cells incubated for up to 28 days under air-liquid-interface (ALI) reached very high transepithelial electrical resistance (TEER) of approximately 6000 Ω *cm² (Fig. 1b). In the revised manuscript we added investigations on TEER values of SARS-CoV-2 D614G infected and mock-treated hAELVi ALI cultures. This analysis revealed a decrease of the TEER in virus-infected samples by about 1.500 Ω *cm² until 72 h p.i.. This finding suggests that barrier properties of the cellular monolayer are basically maintained, but impaired by virus infection. Here, we focused on the impact of SARS-CoV-2 infection on TEER values, because parallel measurements of cultures infected with other viruses such as IAV, are hampered by practical reasons, like an obligatory sterilization step of the technical equipment that would had been required between measurements for several viruses. Data on development of TEER values upon infection are depicted in the new supplementary figure 3 and needed addition of lines 120-122 in the results. We are aware that analyses of protein levels or phosphorylation states of proteins related to tight junctions in virus-infected hAELVi ALI cultures could be used to further evaluate and extend findings from the TEER measurements (Reiche and Huber, 2020). However, we feel that such analyses are beyond the scope of the current study that had the main goal to describe the use of polarized hAELVi cultures for the phenotypic characterization of respiratory viruses, including SARS-CoV-2.

The fact that this cell line has already been described and generally characterized plus the limited virological studies, weakens the manuscript and the conclusion drawn by the authors.

As detailed in the manuscript, hAELVi cell cultures have been originally established to model essential features of the air-blood barrier of the human lower respiratory tract. Studies using this model system (Joelsson et al., 2020; Leibrock et al., 2019; Mills-Goodlet et al., 2020) focused on absorption and toxicity of inhaled drugs, chemicals and nanomaterials. However, as pointed out, there is no report on their use in infectious diseases or virus research. Possibly, this gap exists because others have failed to establish suitable conditions enabling studies of virus infections in hAELVi cultures. Setting up of a

stable procedure (e.g. culture medium, harvesting of viruses) that supports propagation of different respiratory viruses to high levels, is actually not trivial.

We believe that the characterization of hAELVi cells as infection model to study respiratory virus infections in the lower respiratory tract (LRT) is of particular interest as these cells have numerous advantages compared to other cell lines commonly used as infection models. In contrast to other commonly used human respiratory cell lines, hAELVi cells originate from healthy human lung cells (Kuehn et al., 2016). This is a major advantage since cellular processes may be deregulated in cell lines originating from tumor cells. Further, it has been shown, that presence of an air-liquid-interface leads cells to recapitulate the expression patterns of the human respiratory tract more closely than submerged cultures (Pezzulo et al., 2011) Well characterized human alveolar cancer cell lines like A549 and NCI-H1299 cells do not form tight junctions and are therefore unsuitable for establishing ALI cultures (Hiemstra et al., 2018; Kuehn et al., 2016). In fact, human respiratory cell lines known to polarize under cell culture conditions, like Calu-3, NCI-H441 and 16HBE14 cells, as well as commercially available human airway epithelial cultures are of bronchial origin belonging to the conducting airways and are therefore unsuitable to resemble the distal lung.

Our revised manuscript highlights the high permissiveness of hAELVi ALI cultures for several respiratory viruses including highly pathogenic coronaviruses as well as influenza A virus. Furthermore, we provide insights into virus-triggered innate immune activation and visualize important intracellular aspects of the viral replication cycle of SARS-CoV-2 by ultrastructural analyses. Also, we extended our analyses during revision towards comparisons of related SARS-CoV-2 variants of concern showing that the culture system is capable to discriminate between the growth phenotypes even of closely related viruses. Hence, we feel that this first report on hAELVi cultures as a suitable infection model of the LRT of humans, will not only enable follow-up studies with the goal to nail-down molecular factors associated with high- or low viral virulence, but also provides an opportunity for risk assessment of emerging pathogens and timely information of the Public Health sector.

Minor

The model is not novel; was described by Kuehn A et. al ALTEX 33: 251-60 2016 as cited by the authors of this manuscript.

We did acknowledge in the introduction of the original manuscript that hAELVi cultures had been used for different purposes outside infectious disease. Establishment of experimental conditions (e.g. culture medium, harvesting of viruses) and a protocol that enables viral propagation to high levels, was actually not trivial. Here, we highlighted that using hAELVi cells to investigate respiratory virus infection is a novel and very promising exploitation of this human model. Please also see our reply to the last main topic of referee #2.

With BA.1 and BA.2, it is important for the authors to indicate in the text which Omicron variant is presented in this manuscript.

We totally agree that indication of Omicron sublineages is necessary. However, when submitting the initial manuscript, this differentiation did not exist. We therefore added lineage designations to the respective BA.1 or BA.2 isolates throughout the revised manuscript.

The authors cite Peacock et. al but don't discuss the fact that the Peacock manuscript demonstrated better Omicron replication in nasal derived cells. This point between the two different cell types and replication ability may demonstrate differences between transmission and pathogenicity of variants and should be mentioned in some degree.

We extended discussion about the study of Peacock et al. in the revised manuscript, thereby pointing out that high transmissibility of Omicron VOC may be due to its enhanced replication in nasal cells (lines 192-194 and 198-202).

References

Halfmann, P.J., Iida, S., Iwatsuki-Horimoto, K., Maemura, T., Kiso, M., Scheaffer, S.M., Darling, T.L., Joshi, A., Loeber, S., Singh, G., *et al.* (2022). SARS-CoV-2 Omicron virus causes attenuated disease in mice and hamsters. *Nature*.

Hiemstra, P.S., Grootaers, G., van der Does, A.M., Krul, C.A.M., and Kooter, I.M. (2018). Human lung epithelial cell cultures for analysis of inhaled toxicants: Lessons learned and future directions. *Toxicol In Vitro* 47, 137-146.

Joelsson, J.P., Myszor, I.T., Sigurdsson, S., Lehmann, F., Page, C.P., Gudmundsson, G.H., Gudjonsson, T., and Karason, S. (2020). Azithromycin has lung barrier protective effects in a cell model mimicking ventilator-induced lung injury. *Altex* 37, 545-560.

Kuehn, A., Kletting, S., de Souza Carvalho-Wodarz, C., Repnik, U., Griffiths, G., Fischer, U., Meese, E., Huwer, H., Wirth, D., May, T., *et al.* (2016). Human alveolar epithelial cells expressing tight junctions to model the air-blood barrier. *ALTEX* 33, 251-260.

Leibrock, L., Wagener, S., Singh, A.V., Laux, P., and Luch, A. (2019). Nanoparticle induced barrier function assessment at liquid-liquid and air-liquid interface in novel human lung epithelia cell lines. *Toxicol Res (Camb)* 8, 1016-1027.

Mills-Goodlet, R., Schenck, M., Chary, A., Geppert, M., Serchi, T., Hofer, S., Hofstätter, N., Feinle, A., Hüsing, N., Gutleb, A.C., *et al.* (2020). Biological effects of allergen–nanoparticle conjugates: uptake and immune effects determined on hAELVi cells under submerged vs. air–liquid interface conditions. *Environmental Science: Nano* 7, 2073-2086.

Peacock, T.P., Brown, J.C., Zhou, J., Thakur, N., Newman, J., Kugathasan, R., Sukhova, K., Kaforou, M., Bailey, D., and Barclay, W.S. (2022). The SARS-CoV-2 variant, Omicron, shows rapid replication in human primary nasal epithelial cultures and efficiently uses the endosomal route of entry.

Pezzulo, A.A., Starner, T.D., Scheetz, T.E., Traver, G.L., Tilley, A.E., Harvey, B.G., Crystal, R.G., McCray, P.B., Jr., and Zabner, J. (2011). The air-liquid interface and use of primary cell cultures are important to recapitulate the transcriptional profile of in vivo airway epithelia. *Am J Physiol Lung Cell Mol Physiol* 300, L25-31.

Reiche, J., and Huber, O. (2020). Post-translational modifications of tight junction transmembrane proteins and their direct effect on barrier function. *Biochimica et biophysica acta Biomembranes* 1862, 183330.

Schmidt, F., Weisblum, Y., Muecksch, F., Hoffmann, H.H., Michailidis, E., Lorenzi, J.C.C., Mendoza, P., Rutkowska, M., Bednarski, E., Gaebler, C., *et al.* (2020). Measuring SARS-CoV-2 neutralizing antibody activity using pseudotyped and chimeric viruses. *J Exp Med* 217.

Uraki, R., Kiso, M., Iida, S., Imai, M., Takashita, E., Kuroda, M., Halfmann, P.J., Loeber, S., Maemura, T., Yamayoshi, S., *et al.* (2022). Characterization and antiviral susceptibility of SARS-CoV-2 Omicron BA.2. *Nature* 607, 119-127.

Willett, B.J., Grove, J., MacLean, O.A., Wilkie, C., De Lorenzo, G., Furnon, W., Cantoni, D., Scott, S., Logan, N., Ashraf, S., *et al.* (2022). SARS-CoV-2 Omicron is an immune escape variant with an altered cell entry pathway. *Nat Microbiol*.

REVIEWERS' COMMENTS:

Reviewer #1 (Remarks to the Author):

I thank the authors for their responses to my comments and I am satisfied that these have all been appropriately addressed. Further, I commend them on a well-presented, interesting study.

I have a few extra minor formatting edits for them to consider

1. Figure 2 line 483, I believe there is an error regarding the panel labelling.
2. Figure 3 panel C, the labelling for the BA.1/2 should be consistent with the panel A/B, i.e. either 'SARS-CoV-2 Omicron BA.1' or SARS-CoV-2 BA.1'.
3. When referring to measurements performed on apical washes of ALI cultures in the text I think the authors should refrain from referring to these as 'supernatants' (e.g. line 139) and instead describe these as apical washes for the sake of clarity for the reader.

Reviewer #2 (Remarks to the Author):

My comments and concerns were addressed by the authors in the latest version of the manuscript.